# Imaging with spatio-temporal modelling to characterize the dynamics of plant-pathogen lesions

**Melen Leclerc**[1]*, **Stéphane Jumel**[1], **Frédéric M. Hamelin**[1], **Rémi Treilhaud**[1], **Nicolas Parisey**[1], **Youcef Mammeri**[2]

**1** IGEPP, INRAE, Institut Agro, University of Rennes, Rennes, France, **2** ICJ, CNRS, Jean Monnet University, Saint-Etienne, France

* melen.leclerc@inrae.fr

## Abstract

Within-host spread of pathogens is an important process for the study of plant-pathogen interactions. However, the development of plant-pathogen lesions remains practically difficult to characterize beyond the common traits such as lesion area. Here, we address this question by combining image-based phenotyping with mathematical modelling. We consider the spread of *Peyronellaea pinodes* on pea stipules that were monitored daily with visible imaging. We assume that pathogen propagation on host-tissues can be described by the Fisher-KPP model where lesion spread depends on both a logistic growth and an homogeneous diffusion. Model parameters are estimated using a variational data assimilation approach on sets of registered images. This modelling framework is used to compare the spread of an aggressive isolate on two pea cultivars with contrasted levels of partial resistance. We show that the expected slower spread on the most resistant cultivar is actually due to a significantly lower diffusion coefficient. This study shows that combining imaging with spatial mechanistic models can offer a mean to disentangle some processes involved in host-pathogen interactions and further development may allow a better identification of quantitative traits thereafter used in genetics and ecological studies.

## Author summary

The study of plant diseases often rely on the measurement of quantitative traits that describe the development of a pathogen into a host plant. When the pathogen causes growing lesions, phenotyping is actually more difficult and usually summarized into lesions sizes. By considering the spread of *Peyronellaea pinodes* in two pea cultivars with contrasted level of quantitative resistance we show how visible image-based phenotyping combined with a spatial mathematical model can improve the phenotyping of interactions with growing lesions. It provides new life-history traits of the pathogen that cannot be identified without using spatial models and better describe pathogen spread into host tissues. The comparison of these quantitative traits among cultivars provides better insight into the possible mechanisms involved in quantitative host resistance. Our results point

**Data Availability Statement:** The authors confirm that all data underlying the findings are fully available without restriction. All relevant data are

within the paper and its Supporting information files.

**Funding:** This study was funded by the Bretagne Loire University and the Plant Health and Environment Division of INRAE through StartIRM (ML) and MODIM projects (ML). The funders had no role in study design, data collection and analysis, decision to publish, or preparation of the manuscript.

**Competing interests:** The authors have declared that no competing interests exist.

out the relevance of going further than usual traits like lesion size or lesion growth rate to characterize diseases with growing lesions and support the idea of promoting model-based phenotyping in works aiming to understand the adaptation of pathogen to plant resistances and breed for resistance.

## Introduction

Assessing quantitative life-history traits of pathogens on host plants, also called quantitative traits of pathogenicity or aggressiveness by plant pathologists, is central for the study of plant diseases. These quantitative traits describe the essential stages of pathogens life-cycle into their hosts and are used to understand the adaptation of pathogens to plant resistance and to identify quantitative trait loci for both host resistance and pathogen aggressiveness [1]. In the particular case of fungal plant pathogens, the most frequently measured traits are incubation and latency periods, spore production and lesion size [2]. In practice, they are often obtained after inoculating a host, monitoring the development of the lesions caused by the pathogen and finally estimating the traits of interest. However, phenotyping the dynamics of host-pathogen interactions remains challenging and is often performed through inaccurate traits that, though they already contrast phenotypes, poorly describe the processes and can hide or skew differences between individuals [3]. The lesion size is a good example to illustrate this as there is an infinity of spatial dynamics than can produce identical size at a given time. Considering the lesion growth rate is more informative but again, it ignores lesions shapes and depends on processes such as local growth and diffusion. Mechanistic models offer a mean to decipher the processes involved in host-pathogen interactions but are still seldom considered for analyzing plant disease phenotypic data [3, 4]. In this case the model should remain parsimonious enough so that the parameters can be identified from the data.

The recent development of image-based phenotyping methods enables *in vivo* non-destructive longitudinal monitoring of infected tissues. Besides allowing precise and automated quantification of necrotic plant tissues, that already improved disease phenotyping (e.g. [5–7]), imaging opens new possibilities to further investigate the spatial dimension of host-pathogen interactions. As illustrated by works on the development of human lesions, imaging data can be particularly interesting for fitting spatially explicit process-based models [8, 9]. It provides new insights into the main mechanisms involved in lesion development in relation with host immunity but also modelling tools for phenotyping. Perhaps surprisingly, although the main physiological mechanisms of plants and their parasites have been described by mathematical models (e.g. [10, 11]) the spread of lesions has received little attention by modellers [3, 12–14] and is rarely validated against images [15].

In this study we consider the fungal pathogen *Peyronellaea pinodes* (formerly *Mycosphaerella pinodes* and *Didymella pinodes*) on pea as an example pathosystem to analyze its spread using modelling and imaging. With the two fungi *Phoma medicaginis* and *Ascochyta pisi*, *P. pinodes* belongs to the Ascochyta blight of pea disease complex that causes substantial yield losses worldwide [16]. In Europe, *P. pinodes* is generally the predominant and the most destructive species, though *P. medicaginis* is also prevalent and tends to develop later in the growing season [17, 18]. *P. pinodes* is able to infect all aerial parts of its host plant and induces necrotic growing lesions. The development of resistant cultivars to *P. pinodes* has been central for the integrated management of this disease, but difficult as only quantitative (or partial) resistance was available [16, 19]. For most fungal pathogens, quantitative host resistance reduces pathogen fitness by altering spore production, infection and within-host growth [1–3,

20]. The evaluation of quantitative host resistance on pathogen life-history traits is often performed in controlled conditions with *ad hoc* protocols. In the case of *P. pinodes* partial resistance of pea can be assessed on inoculated detached leaflets, or stipules, by measuring necrotic lesions either manually [21] or with imaging [17].

We begin by presenting the experiment, including the image acquisition protocol and the processing framework, that allowed the longitudinal monitoring of lesions on inoculated pea stipules. Then, we consider the Fisher-KPP reaction-diffusion model to describe the spread of necrotic lesions on host tissues which is fitted to image sequences (Fig 1). We show that combining imaging and spatially explicit models enables a finer description of within-host spread of pathogen, including lesions coalescence, and allows one to disentangle local growth and diffusion of the necrosis. The comparison of estimated parameters obtained on two cultivars provides new insights into how the partially resistant cultivar reduces pathogen development. It highlights the potential of combining image-based phenotyping with spatial mechanistic models to further improve our understanding of host-pathogen interactions.

## Materials and methods

### Host inoculation experiment

The aggressive isolate of *P. pinodes* named *Mp 91.31.12* was inoculated on two pea cultivars previously tested in our laboratory: Solara, a common susceptible reference, and James, that reduces symptom development in controlled conditions [21]. These two cultivars are semi-leafless without conventional leaves but extended pairs of hypertrophied stipules below each stem node. For each cultivar, leaf stipules were inoculated according to a standard biotest protocol developed in our laboratory [17, 21]. Plants were grown in a climate chamber, kept at 18˚C and with a 12h photoperiod, in 9cm diameter pots containing vermiculite and five pea seeds. When they reached the 6 leaf stage, stipules from nodes 3 and 4 were sampled and placed on tap water in a compartmented square Petri dish. The inoculum consisted in a pycni-diospore suspension whose concentration was determined with a haemocytometer and adjusted at $5 \times 10^4$ spores ml$^{-1}$ following [21] protocol. For both cultivar, 16 pairs of attached stipules were inoculated by placing a 10$\mu$l droplet at their center. Afterwards, the Petri dishes containing the inoculated stipules were placed into transparent plastic containers to avoid drop evaporation and incubated in a climatic chamber kept at 20˚C and with a 14 h photoperiod. The protocol is summarized into a schematic diagram given in S1 Appendix.

### Image acquisition

The spread of lesions caused by the pathogen was assessed daily from 3 to 7 days after inoculation following a standardized acquisition protocol developed for plant disease phenotyping [6, 17, 22]. Image acquisition was performed using two FotoQuantum LightPro 50 × 70 cm soft boxes, placed on both sides of the Petri dish with four daylight bulbs each (5400 K, 30 W). Pictures were taken with a Nikon D5300 digital camera equipped with an AF-S DX Micro Nikkor 40 mm 1:2.8G lens, on a Kaiser Repro stand, and with computer control using DigiCamControl software ver. 2.1.1.0. Aperture was set at F22 for maximal depth of field, iso 125, daylight white balance. Initial pictures were saved as RGB images with a resolution of 6000 × 4000 pixels.

### Image processing

As illustrated in Fig 1, several processing steps were required to enable model fitting to image sequences. First, stipules (i.e. our region of interest) were extracted from raw images using the

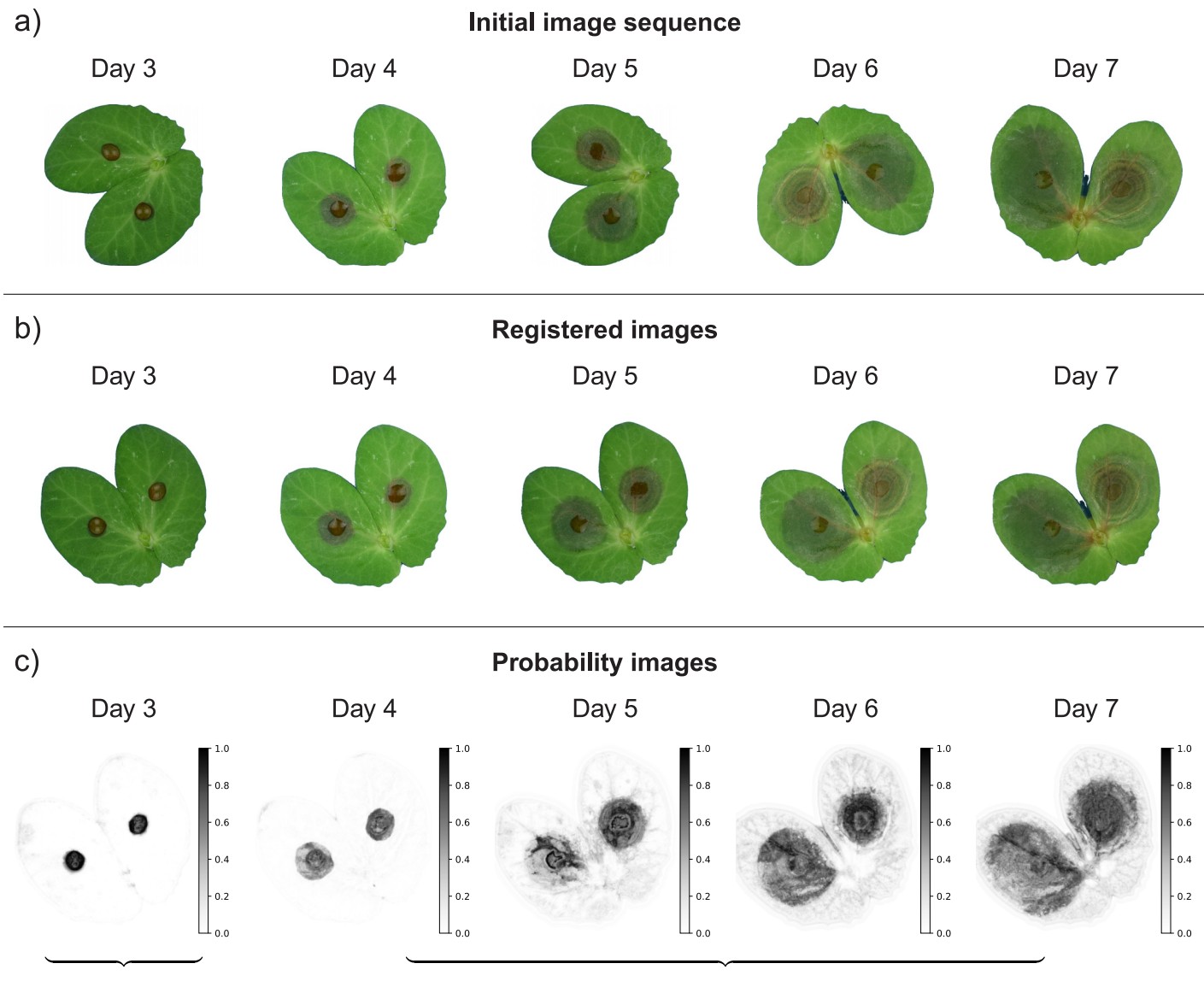

**Fig 1. Schematic representation of lesion growth monitoring through imaging.** The initial RGB images (a) are first registered to align stipules in time (b). Afterwards, a supervised segmentation is performed to produce probability maps indicating the probability of each pixel to be in either healthy, symptomatic or background classes. Probability images of the symptomatic state (c) are used for fitting the Fisher-KPP model. Images of day 3 are used as initial conditions while the remaining 4 images are used to estimate the pathogen local growth rate $\hat{a}$ and diffusion coefficient $\hat{D}$ that are actually two distinct life-history traits of within-host pathogen spread.

Simple Interactive Object Extraction algorithm [23] (Fig 1a). At this step we saved images with stipules on a white background instead of stipules masks (i.e. binary images). Second, images were registered (i.e. aligned to each other) using the Coherent Point Drift method [24], assuming rigid transformations and the first image (3 days after inoculation) as the reference (Fig 1b). Third, images were segmented by classifying pixels in either healthy, symptomatic or background states. The prediction of each pixel-class was based on several nonlinear image features that captured local image characteristics. In particular we computed features for

colours (e.g. Gaussian blur), edges (e.g. Laplacian), and textures (e.g. Hessian) at different scales using spherical filters with radii varying from 1 to 16 pixels. Based on these features, Random Forest classifiers were trained for each date of observation using the Trainable Wai-kato Environment for Knowledge Analysis (Weka) [25]. They were tested on ground truth images from an independent study [17] and showed good performances to classify pixels (i.e. balanced accuracies above 0.82, S2 Appendix), and thus separate background, healthy leaf and symptomatic areas. Afterwards, these classifiers were used to process the full dataset and to get three probability images giving the probabilities of each pixel to be in each state (e.g. Fig 1c and S3 Appendix for state symptomatic). The predicted state probabilities correspond to the mean predicted state probabilities of the trees in the forest. For each pixel the sum is 1 and the state (or class) obtained in the pixel-based segmentation is given by the class with the highest probability (soft voting). To finish with, we considered the Jaccard index (i.e. the intersection over the union of the two sets, that can vary between 0 and 1) as a measure of stipules deforma-tion, that was computed for each date assuming the 3$^{rd}$ day as the reference set. All the images and classifiers are available in an open dataverse [26, 27].

## Spatial lesion growth model

Most existing models for the spread of plant pathogens within host tissues are rather spatially implicit and generally assume a constant radial growth rate and a simplified geometry of the host organ [12–14]. Yet, these models were able to fit non-spatial lesion size data [3], including for the particular *P. pinodes*-pea pathosystem [17].

Here, we consider the Fisher-KPP equation as a model for the spatio-temporal dynamics of lesions. Because pathogen density cannot be directly inferred from common observations of symptoms in biotests, we describe the spread of the probability of infection, and thus the appearance of symptomatic host tissues, rather than pathogen load. The Fisher-KPP equation was introduced in 1937 by Fisher [28] and Kolmogorov-Petrovsky-Piskunov [29] as a semi-linear parabolic partial differential equation (PDE) combining Fick's diffusion with logistic growth. Let $\Omega \subset \mathbb{R}^2$ be the stipules area, the Fisher-KPP equation reads as the following reac-tion-diffusion equation, for the position $\mathbf{x} = (x, y) \in \Omega$ and the time $t > t_0$

$$\frac{\partial u}{\partial t}(\mathbf{x}, t) = D\Delta u(\mathbf{x}, t) + au(\mathbf{x}, t)(1 - u(\mathbf{x}, t)). \tag{1}$$

where $u(\mathbf{x}, t)$ the probability that the host is infected at location $\mathbf{x}$ and time $t$, $D > 0$ is the diffu-sion coefficient, $a \geq 0$ the growth rate. As *P. pinodes* is a necrotrophic pathogen, we can assume that the probability of being necrotic is similar to the probability of being infected. The initial conditions are given by an initial image $u_0$ as

$$u(\mathbf{x}, t = t_0) = u_0(\mathbf{x}) \text{ in } \Omega.$$

Assuming the pathogen cannot move out of the leaf, homogeneous Neumann boundary conditions are imposed

$$\frac{\partial u}{\partial n}(\mathbf{x}, t) = 0 \text{ on } \partial\Omega.$$

This model exhibits traveling waves with asymptotic speed $2\sqrt{aD}$ which is coherent with the assumption of a constant radial growth rate considered in several studies and supported by non-spatial lesion data [3, 17].

Numerical solutions of the model are obtained by computing the spatial domain $\Omega$ with a level-set formalism so the boundaries $\partial\Omega$ match those of the leaves in the image [30, 31], and

solving the partial differential equations using explicit Euler finite differences in time and second order centered finite differences in space. More details on these numerical aspects are provided in S4 Appendix.

## Parameters estimation from image sequences

For each inoculated stipule, the observations consisted in a set of registered images $u_{reg}(\mathbf{x}, \mathbf{t})$ for times after inoculation $\mathbf{t} = \{t_3, t_4, t_5, t_6, t_7\}$ (Fig 1). Parameters identification consisted in seeking estimates $\hat{\theta}$ such that the output of the spatial model $u(\mathbf{x}, t, \theta)$ matches these observations. Depending on the estimation problem, inverse problems or statistical inference of reaction-diffusion can be addressed by several methods such as mathematical analysis, maximum likelihood or non-linear least-squares [32]. When observations are image sequences, parameter estimation can be performed using some data assimilation methods used to fit models to image data, for instance in fluid dynamics [33] or biomedical modelling [9]. We consider a variational data assimilation approach based on optimal control theory [34]. The estimation procedure is based on the nonlinear least-squares cost function:

$$J(\theta) = \frac{1}{2} \sum_{t \in \mathbf{t}} \sum_{\mathbf{x} \in \Omega} \left( u(\mathbf{x}, t, \theta) - u_{reg}(\mathbf{x}, t) \right)^2. \qquad (2)$$

Following the variational assimilation framework [34], estimates $\hat{\theta}$ are found by minimizing $J(\theta)$ thanks to the Lagrangian function $\mathcal{L}(\theta)$ and a numerical procedure both detailed in S4 Appendix. This approach enables a more efficient numerical optimization and thus a faster parameter estimation.

In our case, the probability images (Fig 1) were considered as the observation $u_{reg}(\mathbf{x}, \mathbf{t})$ in the cost function Eq (2) for parameters estimation [8]. For each image sequence, the image of the $3^{rd}$ day provides the initial conditions ($t_0 = t_3$ and $u_0(\mathbf{x}) = u_{reg}(\mathbf{x}, t_3)$)) while the remaining 4 images at times $\{t_4, t_5, t_6, t_7\}$ enables stable estimation of parameters [35]. Even if pixel data are spatially and temporally correlated we did not consider any structure of the errors to fit the model to the image data. While accounting for correlated observation errors would improve parameters estimation it remains challenging in image data assimilation [36].

As the diffusion coefficient depends on image size, we rather consider the relative diffusion (in $cm^2.day^{-1}$) obtained from the raw diffusion coefficient and the stipules area previously extracted from the original images with dedicated landmarks used in the acquisition setup. The model was fitted to the $2 \times 16$ inoculated stipules and we compared the two cultivars through one-way ANOVAs on the estimated diffusion coefficient $\hat{D}$ and growth rate $\hat{a}$. The adequacy of the Fisher-KPP model to the data was assessed by comparing observations against fitted models' predictions and by viewing the raw residuals for each date, i.e. $[u_{reg}(\mathbf{x}, t_i) - u(\mathbf{x}, t_i, \hat{\theta})]$ for $t_i \in \mathbf{t}$ (S3 Appendix). In order to evaluate the ability of the model to capture lesions sizes rather than probabilities of infection, we estimated symptomatic surfaces for each image by counting the number of pixels with probabilities higher than 0.5 in probability maps and model outputs, and then compared the estimated values.

A Python code for fitting the Fisher-KPP on image sequences is available in an open repository [37].

## Results

The Fisher-KPP model and its numerical resolution were able to describe the spread and the coalescence of lesions caused by *P. pinodes* on pea stipules as observed in standard biotests (Fig 2, S1 and S2 Movies). The image processing framework associated with the data

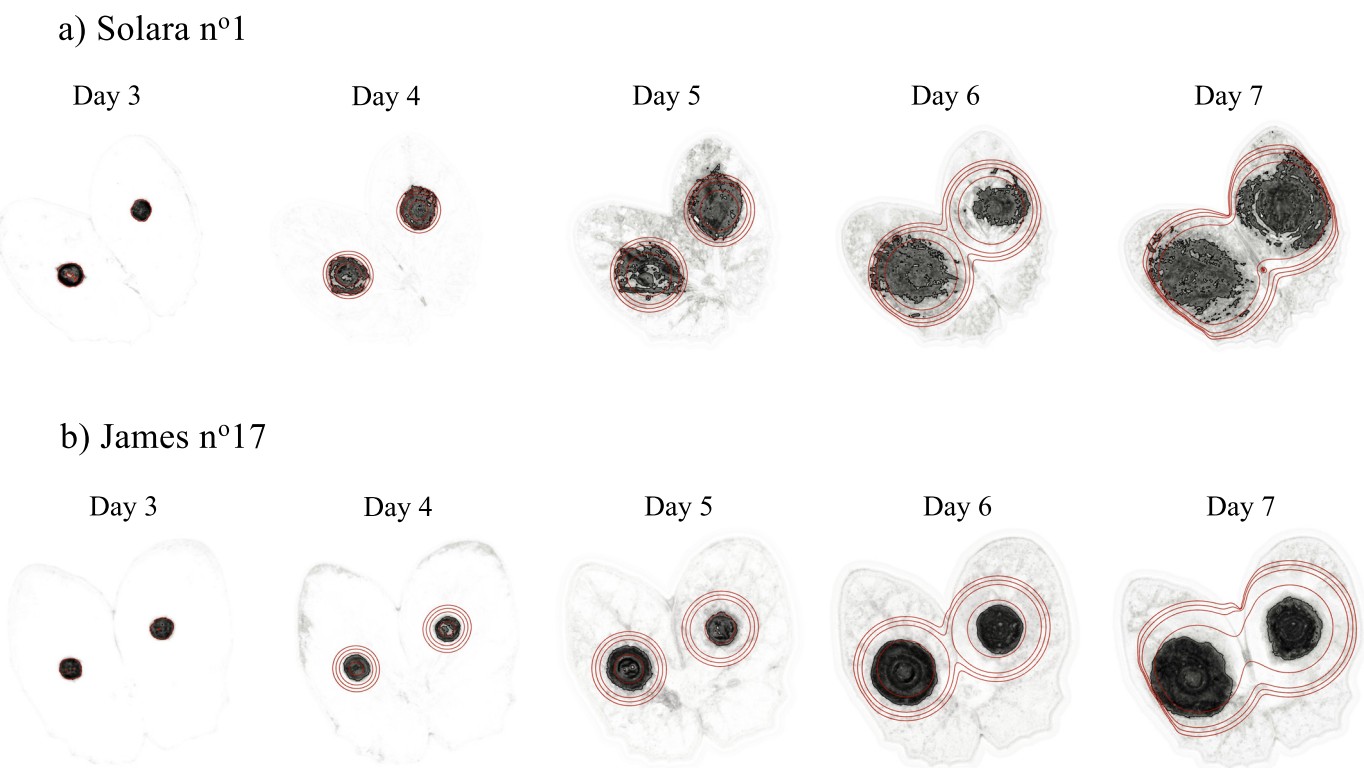

**Fig 2. Visualization of model prediction against image data.** The solution of the fitted Fisher-KPP equation, i.e. with optimal estimated parameters $\hat{\theta}$, is represented through time by contours (0.2, 0.3, 0.4, 0.5) overlying the probability images of the symptomatic class for example stipules of both cultivars, i.e. Solara n˚1 (a) and James n˚17 (b). This comparison between the spatial model and the data is also available in S1 and S2 Movies.

assimilation method allowed us to fit the reaction-diffusion model that captured the essential patterns of the spatio-temporal data (S3 Appendix). Except for one stipule (James N˚27), for which only one lesion was detected in the image at day 3, used for initial conditions, the model always described the joint spread of the two lesions. In detail, the comparison between the fitted models and the image data suggests that the model tends to underestimate the probability of infection, especially at days 5 and 6. The discrepancy between the reaction-diffusion model and the data may be mostly explained by patterns observed in the probability images (S3 Appendix). Even if the trained Random Forest classifiers showed good performances to classify pixels (S2 Appendix), classification became less certain over time with lower probabilities (Fig 1c). While this may not impact pixel classification through soft voting (the probability of the symptomatic class is still the highest), it can introduce some noise and residual errors in mechanistic model fitting. At days 5 and 6 the contours of the lesions are relatively easy to draw and the segmentation exhibits high probabilities in the symptomatic areas. On the contrary for the last two days the appearance of the lesions changes and it becomes more difficult for an expert (or annotator), and for an algorithm, to separate healthy and symptomatic parts of the stipules. As a consequence, within lesions some pixels had lower probabilities of being symptomatic on the last dates than for the first ones which induced discrepancy with the Fisher-KPP model (S3 Appendix). Moreover, in this study we assumed a fixed (non-moving or deforming) domain for the reaction diffusion using the leaf contours in the first image (day 3). Although stipules deformation remained limited (S5 Appendix) with Jaccard indexes above

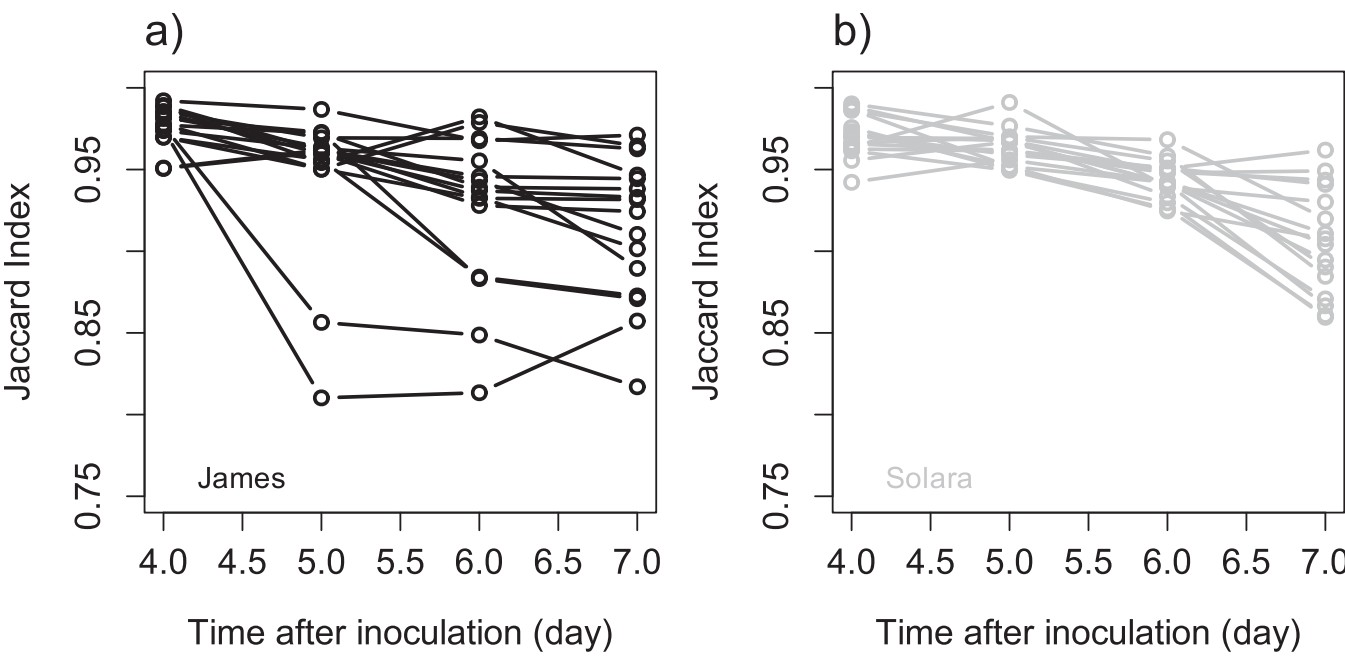

**Fig 3. Visualization of stipules deformation in time.** Change in the Jaccard index with time for cultivars James (a) and Solara (b). At each time after inoculation the Jaccard index was calculated in comparison with the image at day 3, also used as a reference for image registration.

0.8 for all individuals (Fig 3), it occurred and could have induced some errors between the PDE model and images as illustrated in Fig 2b at day 7. To finish with, when looking at symptomatic surface instead of infection probability, the model exhibited a better adequacy with the image data (i.e. relationship close to the first bisector (Fig 4)). Overall, despite some discrepancy between the model and the probability images, the Fisher-KPP model appeared to be relevant for describing the evolution of symptomatic areas.

We successfully estimated the growth rate $\hat{a}$ and the diffusion coefficient $\hat{D}$ for the 32 monitored individuals (S6 Appendix). James cultivar was characterized by smaller stipules than Solara with average surfaces of 4.49 and 7.93 cm$^2$ respectively. The spread of the pathogen on James cultivar was characterized by mean growth rate $\hat{a}$ and diffusion coefficient $\hat{D}$ of respectively 0.55 and 1.33 against 1.26 and 1.54 on the more susceptible Solara. As suggested by the distributions of parameters estimates (Fig 5), analyses of variance pointed out a significant difference between cultivars for the diffusion coefficient $\hat{D}$ (p-values $<0.05$, S6 Appendix) whereas the analysis does not support the hypothesis different growth rates $\hat{a}$ (p-values $> 0.1$, S6 Appendix). These results suggest that the partial resistance of James, previously observed in controlled conditions, may only due to mechanisms that slow down pathogen diffusion into host tissues. Therefore, for inoculated stipules with identical areas and shapes, lesions caused by *P. pinodes* will spread at a higher speed, and thus coalesce and reach edges earlier, on Solara than on James.

## Discussion

In this study we combined image processing and mathematical modelling to investigate the dynamics of host-pathogen interactions. We showed that a longitudinal monitoring of inoculated leaves through visible imaging provides data to fit reaction-diffusion models that describe

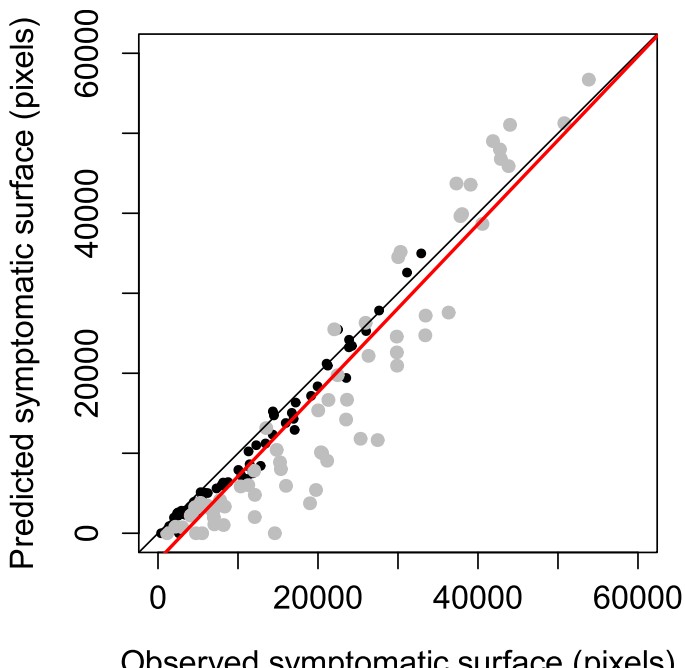

**Fig 4. Comparison of symptomatic surface predicted by the reaction-diffusion models against the surface obtained after pixel-based image segmentation for all images used to fit the model.** For each cultivar, i.e. James (black points) and Solara (gray points), there are thus 16 stipules × 4 dates points. The black line is the first bisector that indicates a perfect agreement between values while the red line is the estimated linear relationship between prediction and observation considering all data (slope = 1.05, intercept = −3392.0). In detail, the relationship for James cultivar was better (slope = 1.01, intercept = −1418.7) than for James and (slope = 1.14, intercept = −6947.1).

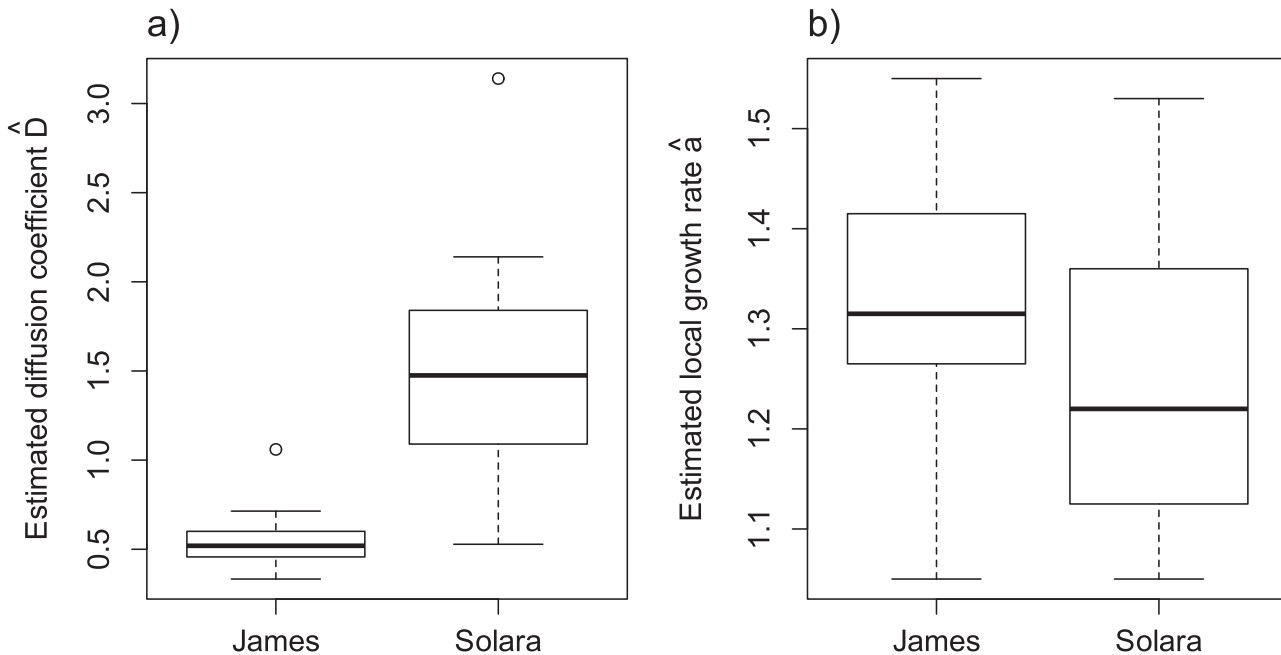

**Fig 5. Distributions of the estimated parameters.** a) diffusion coefficient $\hat{D}$ with a mean values of 0.55 for James against 1.54 for the more susceptible cultivar Solara, b) local growth rate $\hat{a}$ with mean values of 1.33 and 1.26 for respectively James and Solara. The 32 estimated parameters (16 for each cultivar) are available in S6 Appendix.

the spatio-temporal spread of pathogen on host tissues. While such methodological approaches are common in biomedical sciences (e.g. [9, 38]) they are original in plant pathology. Here, we considered the fungal pathogen *P. pinodes* on pea as an example pathosystem and used the Fisher-KPP equation to model necrotrophic lesions. Using this PDE model with a variational data assimilation method we were able to capture the essential patterns of image-sequences data and disentangle growth and diffusion. These processes are actually two distinct life-history traits that both explain host colonization by the pathogen through lesions. They provide a finer description of the interaction but cannot be determined without the use of spatially-explicit models with spatial information, as provided by images, because different growth rates and diffusion coefficients can lead to identical lesion speed. Furthermore, while lesions coalescence, different leaves sizes, or lesion saturation at leaves boundaries can be problematic when comparing lesion sizes in common aggressiveness biotests, the inference of parameters in parsimonious PDE models can handle properly such situations.

We assessed the development of an aggressive isolate on two cultivars with contrasted level of partial resistance using a standard protocol developed for screening both pathogen aggressiveness and host resistance [21]. Our results were consistent with previous findings as the spread of lesions caused by *P. pinodes* was slower on James than Solara. Most interestingly, by combining image-based phenotyping and mechanistic modelling we found that the partial resistance of James were explained by a significant lower diffusion coefficient and not by a decreased growth rate. Although this finding may not be a posteriori surprising it remains impossible to demonstrate using usual measurements of lesions size without the use of spatial models and data. The diffusion coefficient and the local growth rate could be considered as hard traits that capture the function of interest but are difficult to measure [39]. On the opposite, lesion size (or lesion growth rate) are soft traits that are easier to measure and are surrogates of the hard traits [1]. Hard traits obtained with the use of mechanistic models provide a better description of processes involved in pathogen fitness [3, 20]. Although our results are obtained on only two cultivars, this study demonstrates the potential of combining imaging with mathematical models to improve the comparison of cultivars (or isolates) and gain new insights into plant resistance to disease. Optical sensors recently percolated in plant sciences and contributed to recent development of precision phenotyping for plant diseases [5, 7, 40]. On the other side the usefulness of mechanistic models for analyzing phenotypic data is recognized (e.g. [3, 4]) but remains seldom considered. We think that the combination of these two approaches would be particularly relevant for investigating host-pathogen interactions in relation with quantitative or partial resistance.

Although our modelling framework was able to describe the overall visible spread of necrotrophic lesions caused by *P. pinodes* it may be improved and extended on several points. Firstly, for the sake of simplicity we ignored stipules deformation. This change in the shape of host organs caused by parasitism frequently occurs in plants and would be worth considering using existing mathematical and numerical methods for explicit modelling of shapes [30, 41, 42] or plant growth [43]. While such improvement would increase the complexity of the model, it may contribute to decrease the discrepancy between the model and the data, and perhaps, help to identify genotypes that are less susceptible to disease-induced deformation. Secondly, the inoculated host leaves were digitalized through visible imaging and the reaction-diffusion model was fitted to probability images obtained with trained classifiers [8]. The appearance models, learned by experts, that transform raw image into an output which match with the state variable of the process model can have an influence on parameter estimation. In our case we could improve the classifiers to reduce the noise that occur in time by training more advanced algorithm for pixel-based segmentation [44] or include some filtering after predictions (e.g. morphological closing). Comparing different appearance models or

segmentation algorithms and assessing how they modify parameters estimation would be interesting, especially for pathosystems that cause unclear lesions that are difficult to annotate. For practical reasons we ignored any spatial and temporal correlation in pixel data for parameters estimation. As such a strong assumption can bias parameters estimates (mean and variance) it would be worth considering correlated errors in future works [36]. Yet, for comparing cultivars and isolates we believe that the bias introduced by the classifier might have a stronger impact on parameters than the assumption of uncorrelated data. Moreover, because there is no direct relationship between the appearance of symptoms and pathogen density in infected tissues we rather considered the spread of the probability of infection. Although this choice and the interpretation of some parameters could be criticized, it seems to describe well enough the dynamics of infections. In our case, whereas the estimated diffusion coefficient may describe well enough within-host spread we believe that the growth rate should be taken with care. In further studies it would be very interesting to assess the spread of pathogen density using destructive sampling with real-time quantitative PCR [45, 46] and non-destructive monitoring, e.g. with bioluminescence imaging [47]. Thirdly, as lesions caused by *P. pinodes* appeared to spread at a constant speed with quite homogeneous patterns, we choose the Fisher-KPP equation. Although this first model already described the essential patterns of the data, it would be interesting to relax some of its assumptions to improve the description of the observed spatial dynamics. For instance, one could consider a heterogeneous diffusion to capture the acceleration that seems to occur at the end of the experiment. Moreover, the Fisher-KPP equation will not be appropriate to describe the spreading processes that occur in all pathosystems. The spatial dynamics of plant-pathogen lesions remains poorly addressed and further works could benefit from theoretical knowledge on PDE for propagating systems and existing models for the spread of invasive organisms [48, 49], microbial populations and fungal colonies [10], or human lesions [8, 9]. For example, the effect of leaf veins that can guide lesion spread in some pathosystems could be considered through advection terms or by considering hybrid reaction-diffusion models with different dynamics on host tissues (2D) and veins (1D) [49]. On the other side, like microbial populations in controlled media, plant-pathogen lesions can be an interesting experimental systems to test and feed some mathematical theories [50, 51]. Fourthly, our model ignored any host response to infection and further development could take into account some key physiological and immune processes. For instance it would be worth including ontogenetic and disease-induced changes in host susceptibility, e.g. caused by senescence or hypersensitive responses, that are known to occur in several pathosystems and can be spatially localized on leaf tissues [52, 53]. Such phenomena may be taken into account using age-structured PDE considering the age of infection [54] and the physiological age of host tissues.

From an epidemiological point of view the within-host dynamics of the pathogen is an important phase that can have strong impact on epidemics at the population level. Scaling-up the behaviour of epidemics from individuals to populations is still a challenging question for mathematical and computational epidemiology and, at least in the case of plant diseases, the within-host spread of pathogens is either ignored or extremely simplified compared to other epidemiological processes such as spores production [12, 14, 48, 55]. This is mainly due to the challenges of multiscale and spatial modelling, but perhaps, also to the lack of spatial models for within-host pathogen development. Thus, we believe that besides providing new fundamental knowledge and phenotyping tools, spatial lesions models that describe observable spread of pathogen on host organs would also contribute to improve modelling works focused on higher scales. In addition, new insights into the effects of host resistance on within-host dynamics would also feed models for understanding the durability plant resistance to diseases [56–58]. For instance, the impact of partial resistance on either the diffusion coefficient or the

local growth rate may affect differently pathogen fitness and have contrasted impacts on pathogen invasion, persistence and evolution.

## Supporting information

**S1 Appendix. Appendix for the experimental protocol.**
(PDF)

**S2 Appendix. Assessment of classifiers quality.**
(PDF)

**S3 Appendix. Assessment of the discrepancy between model and data.**
(PDF)

**S4 Appendix. Numerical discretization and resolution.**
(PDF)

**S5 Appendix. Visual assessment of stipules deformation.**
(PDF)

**S6 Appendix. Comparison of estimated parameters.**
(PDF)

**S1 Movie. Model prediction against image data for Solara N1.** Movie showing the solution of the fitted Fisher-KPP equation through time overlying the image sequences of the probability images of the symptomatic class for stipule Solara N1.
(MP4)

**S2 Movie. Model prediction against image data for James N17.** Movie showing the solution of the fitted Fisher-KPP equation through time overlying the image sequences of the probability images of the symptomatic class for stipule James N17.
(MP4)

## Acknowledgments

We are grateful to Alain Baranger, Christophe Le May, Agathe Dutt, Lydia Bousset, Théo Fabien and Tristan Boureau for useful discussions, and Claudine Pasco and Amélie Morin for their help during the experiment.

## Author Contributions

**Conceptualization:** Melen Leclerc, Frédéric M. Hamelin, Youcef Mammeri.

**Data curation:** Melen Leclerc, Stéphane Jumel, Rémi Treilhaud.

**Formal analysis:** Melen Leclerc, Stéphane Jumel, Nicolas Parisey, Youcef Mammeri.

**Investigation:** Rémi Treilhaud.

**Methodology:** Melen Leclerc, Nicolas Parisey, Youcef Mammeri.

**Software:** Youcef Mammeri.

**Supervision:** Melen Leclerc.

**Validation:** Stéphane Jumel, Nicolas Parisey, Youcef Mammeri.

**Visualization:** Melen Leclerc, Stéphane Jumel, Youcef Mammeri.

**Writing – original draft:** Melen Leclerc, Youcef Mammeri.

**Writing – review & editing:** Stéphane Jumel, Frédéric M. Hamelin, Nicolas Parisey.

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
