## [Decision Letter · Decision Letter 0]

21 Mar 2023

Dear Dr Leclerc,

Thank you very much for submitting your manuscript "Imaging with spatio-temporal modelling to characterize the dynamics of plant-pathogen lesions" for consideration at PLOS Computational Biology.

As with all papers reviewed by the journal, your manuscript was reviewed by members of the editorial board and by several independent reviewers. In light of the reviews (below this email), we would like to invite the resubmission of a significantly-revised version that takes into account the reviewers' comments.

Please make sure to directly address Reviewer 2's concern about the biological appropriateness of the model.

We cannot make any decision about publication until we have seen the revised manuscript and your response to the reviewers' comments. Your revised manuscript is also likely to be sent to reviewers for further evaluation.

Sincerely,

Benjamin Althouse

Academic Editor

PLOS Computational Biology

Amber Smith

Section Editor

PLOS Computational Biology

Please make sure to directly address Reviewer 2's concern about the biological appropriateness of the model.

Reviewer's Responses to Questions

**Comments to the Authors:**

Reviewer #1: The manuscript presents a study where the expansion of a disease lesion in a leaf was observed using imaging and image analysis, and subsequently the expansion was modelled with Fisher-KPP model that explicitly includes local growth and diffusion to new area. The main biological claim is that partial resistance in a cultivar (in this particular case) is more governed by hindering invasion (diffusion) than local growth. However, the main scientific advance is the use of temporal imaging and mechanistic modelling to study growth of individual lesions (i.e. the methods), while generalizable knowledge about invasion vs local growth would require more work.

In my opinion this approach (temporal imaging and mechanistic modelling) is generally very welcome and will eventually improve our understanding of within host dynamics significantly.

However, I have hard time believing the suitability of the chosen model to understand the lesion growth dynamics. It is hard to evaluate how well the model really corresponds to the data. Based on the videos and snapshot images, it is clear that the model predicts circular probability-of-infection contours, where in practice part of the contour area is certainly infected and part is certainly healthy - resulting in moderate probability of infection. It seems to me, that the model misses the actual lesion expansion dynamics. Having 60-degree angle where the lesion has expanded and tissue became clearly symptomatic, and 300 degrees of clearly healthy, is not well captured by circular contour of 16.7% infection probability. Hence, I'm not convinced of the biological relevance of the results. I believe that a model that allows for anisotropic expansion is needed for gaining biological understanding of the processes of lesion expansion. Higher time resolution would also potentially make it more convincing - now the dynamics is based on four snap shots and the lesion expansion between each of those is quite significant.

This conclusion is really unfortunate, since I think the idea is really valuable, and definitely something that will lead to deeper understanding of pathogen dynamics. Perhaps the idea on its own already merits publication, but here the models correspondence to biology is unfortunately not convincing to me. I would be happy to be proven wrong.

Minor comments:

LL 285-286: Are there examples of this in literature?

LL 292-293: Are stipules area the same as image size, i.e. is it the image size after extracting the stipules from the background? This could be clarified.

LL 296-297: Refer to S5 Appendix where you show the residuals.

S3 figure: probability plot would be likely more informative as 2D where just the colors are used for showing probability (Excluding height) - especially since it seems to be discrete (or discretized?)

S7 appendix, Anova table S2: "Although the cultivar effect is significant on both D and a it explains 84% of the variance for the diffusion and only 9% for the growth rate." - the p-value is 0.069 for parameter a, which makes the above claim a bit unconventional, although we should not be stuck with a godly threshold of p=0.05. This might be good to note, if some readers think that only p<0.05 is significant.

Reviewer #2: Review provided in a pdf file

**Have the authors made all data and (if applicable) computational code underlying the findings in their manuscript fully available?**

Reviewer #1: None

Reviewer #2: Yes

PLOS authors have the option to publish the peer review history of their article (what does this mean?). If published, this will include your full peer review and any attached files.

Reviewer #1: No

Reviewer #2: No
---

## [Decision Letter · Decision Letter 1]

17 Aug 2023

Dear Dr Leclerc,

Thank you very much for submitting your manuscript "Imaging with spatio-temporal modelling to characterize the dynamics of plant-pathogen lesions" for consideration at PLOS Computational Biology. As with all papers reviewed by the journal, your manuscript was reviewed by members of the editorial board and by several independent reviewers. The reviewers appreciated the attention to an important topic. Based on the reviews, we are likely to accept this manuscript for publication, providing that you modify the manuscript according to the review recommendations.

Sincerely,

Benjamin Althouse

Academic Editor

PLOS Computational Biology

Amber Smith

Section Editor

PLOS Computational Biology

Reviewer's Responses to Questions

**Comments to the Authors:**

Reviewer #2: Review of PCOMPBIOL-D-22-01816 « Imaging with spatio-temporal modelling to characterize the dynamics of plant-pathogen lesions »

The authors did a major effort to answer to our questions and to integrate our comments. In our view this new version of the paper is much more easy to follow and understand, giving some important precisions to the points raised by the reviewers. However, we still have some minor comments.

1)We previously showed our concern about the spatial and temporal correlation in pixel data. In the Material and Methods section of the new version of the MS, the authors highlighted that they neglected this correlation. However, contrarily to our suggestion, they did not discuss the implication of this strong assumption. We still believe that at least the nature of the possible bias in parameter estimates (mean value and/or confident interval) generated by this assumption should be mentioned and discussed as a possible extension/improvement of the method. The reader could ask themselves whether these possible biases are negligible with dataset including tens of thousands of pixels. This is why we suggest to more deeply discuss this issue.

2)The authors updated the text at lines 91-116 for describing image processing. However, regarding one of our previous comment, it is still not clear why some pixels can be classified as “background” at step 3, after having applied the SIOE algorithm, which already extracted the stipules from the image, at step 1. Is the “background” at line 94 different from the “background” at line 98?

3)Although we get the sense of lines 179-183, we think that the computation of the lesion size from probability maps is not easy to understand. We suggest to explain this point in two steps. Firstly, the authors estimated the observed (respectively estimated) lesion size by counting the number of pixel with probabilities higher than 0.5 from probability maps (respectively model outputs), for each image. Secondly, the authors compare these observed and estimate values in Fig. 4. Another question here is whether the authors considered the 32 images (16 images X 2 cultivars) or the 64 images (16 images X 2 stipules X 2 cultivars)? Another question is whether the lesion size was estimated only at day 7th? Finally, the author could consider to distinguish the points in Fig.4 for James and Solara cultivars with different shapes.

4)Line 223 the values of coefficient are (likely) inverted. They should be:

“The spread of the pathogen on James cultivar was characterized by mean growth rate a and diffusion coefficient D of respectively 1.33 and 0.55 against 1.26 and 1.54 on the more susceptible Solara.”

5)At line 224 and Fig.5, the authors should mention (if we understood correctly) that the distributions are based on the 32 values estimated for each cultivar.

6)Appendix 3: why not jittering the outliers in figures S32-34? Moreover, the authors should improve figures’ S1-S31 axes in the appendix 3, they are really difficult to read.

In addition to our previous comments, we would like to raise few additional points:

1)For the readers of PLoS CB, it could be interesting that the authors shortly explain why “it is more relevant to rely on data assimilation method (line 151-152)”.

2)To avoid repetitions from the Image processing section, the authors could consider to rephrase lines 159-163 with only: “In our cases the probability images (Figure 1) were considered as the observation ureg(x,t ) in the cost function.”

3)Figure 2: “stipules of Solara (a) AND (not ET) James (b)”. Also, the authors should mention in the legend that “1” and “17” are plant numbers.

4)At line 120: how is the fit quality of non-spatial model?

5)At line 125, replace with “appearance of symptomatic host tissue” ? (as the model does not consider leaf growth).

6)At line 53, 117 (and in many other parts throughout the paper), the authors state that the model describes the spread of necrotic lesion on host tissue, which is effectively the case. We wonder whether defining u as the “probability that a host is infected” (at line 131), is not somewhat misleading. The author could say that u is the probability of a host at location x at time t to be necrotic as the result of the final step infection process.

**Have the authors made all data and (if applicable) computational code underlying the findings in their manuscript fully available?**

Reviewer #2: Yes

PLOS authors have the option to publish the peer review history of their article (what does this mean?). If published, this will include your full peer review and any attached files.

Reviewer #2: No

Figure Files:

Data Requirements:

Reproducibility:

References:

---

## [Editor Report · Decision Letter 2]

23 Oct 2023

Dear Dr Leclerc,

We are pleased to inform you that your manuscript 'Imaging with spatio-temporal modelling to characterize the dynamics of plant-pathogen lesions' has been provisionally accepted for publication in PLOS Computational Biology.

Best regards,

Benjamin Althouse

Academic Editor

PLOS Computational Biology

Amber Smith

Section Editor

PLOS Computational Biology

---

## [Editor Report · Acceptance letter]

13 Nov 2023

PCOMPBIOL-D-22-01816R2 

Imaging with spatio-temporal modelling to characterize the dynamics of plant-pathogen lesions

Dear Dr Leclerc,

I am pleased to inform you that your manuscript has been formally accepted for publication in PLOS Computational Biology. Your manuscript is now with our production department and you will be notified of the publication date in due course.

With kind regards,

Lilla Horvath
